# Association of *APOE* (rs429358 and rs7412) and *PON1* (Q192R and L55M) Variants with Myocardial Infarction in the Pashtun Ethnic Population of Khyber Pakhtunkhwa, Pakistan

**DOI:** 10.3390/genes14030687

**Published:** 2023-03-10

**Authors:** Naveed Rahman, Asif Jan, Muhammad Saeed, Muhammad Asghar Khan, Zahida Parveen, Javaid Iqbal, Sajid Ali, Waheed Ali Shah, Rani Akbar, Fazli Khuda

**Affiliations:** 1Department of Pharmacy, University of Peshawar, Peshawar 25000, Pakistanasif.research1@gmail.com (A.J.);; 2Department of Pharmacy, Qurtaba University of Science and Technology, Peshawar 25000, Pakistan; 3Cardiology Unit Hayatabad Medical Complex, Peshawar 25000, Pakistan; 4Department of Biochemistry, Abdul Wali Khan University Mardan, Mardan 23200, Pakistan; 5Department of Environmental Sciences, University of Lakki Marwat KP, Marwat 28420, Pakistan; 6Department of Biotechnology, Abdul Wali Khan University Mardan, Mardan 23200, Pakistan; 7Department of Pharmacy, Abdul Wali Khan University Mardan, Mardan 23200, Pakistan

**Keywords:** apolipoprotein E, rs7412, rs429358, polymorphism, single nucleotide polymorphism, coronary artery disease, cardiovascular disease, myocardial Infarction, *Paroxonase 1*, Q192R, L55M, genetic biomarker, variant, whole exome sequencing, Pashtun population, Pakistan

## Abstract

Coronary Artery Diseases (CAD) remains the top among Non-communicable Diseases (NCDs). Variations in *Apolipoprotein E* (*APOE*) and *Paroxonase 1* (*PON1*) have been associated with Myocardial Infarction (MI) in several populations. However, despite the high prevalence of CAD, no such study has been reported in the Pashtun ethnic population of Pakistan. We have conducted a two-stage (i.e., screening and validation) case-control study in which 200 cases and 100 control subjects have been recruited. In the first stage, Whole Exome Sequencing (WES) was used to screen for pathogenic variants of Myocardial Infarction (MI). In the second stage, selected variants of both *APOE* and *PON1* genes (rs7412, rs429358, rs854560, and rs662) were analyzed through MassARRAY genotyping. Risk Allele Frequencies (RAFs) distribution and association of the selected SNPs with MI were determined using the Chi-square test and logistic regression analysis. WES identified a total of 12 sequence variants in *APOE* and 16 in *PON1.* Genotyping results revealed that *APOE* variant rs429358 (ɛ4 allele and ɛ3/ɛ4 genotype) showed significant association in MI patients (OR = 2.11, *p* value = 0.03; 95% CI = 1.25–2.43); whereas no significant difference (*p*˃ 0.05) was observed for rs7412. Similarly, the R allele of *PON1* Q192R (rs662) was significantly associated with cases (OR = 1.353, *p* value = 0.048; 95% CI = 0.959–1.91), with particular mention of RR genotype (OR = 1.523, *p* value = 0.006; 95% CI = 1.087–2.132). Multiple logistic regression analysis showed that rs429358 (C allele) and rs662 (R allele) have a significantly higher risk of MI after adjustment for the conventional risk factors. Our study findings suggested that the rs429358 variant of APOE and PON1 Q192R are associated with MI susceptibility in the Pashtun ethnic population of Pakistan.

## 1. Introduction

Non-communicable diseases (NCDs) are becoming the leading cause of mortality, disability, decreased quality of life, and growing healthcare expenses throughout the world [1]. The global mortality rate from these causes is double that of infectious illnesses, and nutritional deficiencies combined [2]. The most frequent among them include cardiovascular diseases (CVDs) are diabetes, malignancies, and chronic respiratory disorders [3]. They are the major cause of mortality in industrialized countries and the second largest cause in developing and underdeveloped nations, accounting for about 74.4% of all fatalities in 2019, which increased by 20.5% from 2009 to 2019 [4].

Coronary Artery Disease (CAD) is a class of CVD which include Myocardial Infarction (MI), hypertension (HTN), and congenital heart disease. Pathology defines MI as myocardial cell loss caused by persistent ischemia [5]. MI is well-known throughout the world for its high mortality and disability rates and is one of the top 10 causes of death in Pakistan [2,6]. There were 4 million MI-related fatalities between 1999 and 2019 [7].

Lifestyle, environmental, and genetic factors have a key role in the development of CVD [8]. Among the former, some major factors to mention are physical activity, diet, smoking habits, obesity, etc. About the latter, information from different studies has recommended a 40–80% genetic link [9,10]. A person with parental history of premature atherosclerosis has a 1.5- to 2-fold risk of developing the same [11]. Risk can easily be predicted through a better understanding of the genetic components [12,13]. In the last decades, human genetics approaches have recognized genes that have possible contributions towards developing MI [14]. Some of these include variants of different genes such as *Apolipoprotein* (*APOE*), *Paroxonase 1* (*PON1*), *Cytochrome P_4501_A_1_* (*CYP_1_A_1_*), *interleukin-6*, *Cholesteryl Ester Transfer Protein* (*CETP*) and many others [15,16,17,18,19].

One of the major cause of the development of CAD is Atherosclerosis which is a pathological procedure in which lipid is accumulated in the intima and media of the blood vessel and thus leads to the formation of plaques [20]. Both youth and adolescents may develop coronary atherosclerosis [21]. Abnormalities in two proteins, namely *APOE* and human *PON1*, play an important role in its development [22,23]. APOE is a serum glycoprotein that plays an important role in the transport and metabolism of lipids and is encoded by the *APOE* gene, which is located on chromosome 19. Exon 4 of the gene has two common SNPs: rs429358 (388 T > C) and rs7412 (526 C > T). Moreover, three alleles (ɛ2(388 T–526 T), ɛ3(388 T-526C), ɛ4(388C-526C)) and six genotypes (ɛ2/ɛ2, ɛ2/ɛ3, ɛ2/ɛ4, ɛ3/ɛ3, ɛ3/ɛ4 and ɛ4/ɛ4) can be formed by the two SNPs [24,25]. Since allele 3 is the most prevalent in populations, it is referred to as “wild-type.”The alleles 2 and 4 are considered variants [26]. Various studies have reported their association with MI in different ethnicities such as Chinese and Russian etc. [27,28]. Similarly, PON1 is a membrane-bound glycoprotein encoded by the *PON1* gene that is located on chromosome 7q21.3-q22.1 [29,30]. It is associated with highly-density lipoprotein (HDL) and is found in a variety of tissues but is predominantly synthesized in the liver [31]. It inhibits the concentration of low-density lipoprotein cholesterol (LDL-C) by hydrolysis of lipid peroxides [31]. PON1 has considerable anti-inflammatory and anti-oxidative actions through its enzymatic Paroxonase, lactonase, and esterase activities [29]. There is some evidence of low serum PON1 activity in patients with lipid disorders such as diabetes mellitus (DM), MI, atherosclerosis, and familial hypercholesterolemia [30]. *PON1* gene has two common polymorphisms, namely L55M and Q192R, of which L55L and Q192Q are regarded as wild type, and Q192R, R192R, L55M, and M55M are considered variant genotypes [8,32,33].

To the best of our knowledge, no such study hasreported the association of these variants in the Pashtun population of Khyber Pakhtunkhwa (KP), Pakistan, despite the reports of increasing incidence of CAD in recent years [34,35]. Owing to their unique cultural practices, social values, lifestyle, and behaviors make them suitable for such studies [34,36]. Considering the importance of the above-mentioned gene variants, it seems suitable to know their association with MI in the said population.

Therefore, this case-control study has been designed to investigate the possible association of *APOE* and *PON1* variants with the risk of MI in the Pashtun ethnic population of KP, Pakistan.

## 2. Materials and Methods

### 2.1. Ethics Statement

Ethical approval was obtained from the Ethical Committee of the Department of Pharmacy, University of Peshawar (No: 906/Pham). Written informed consent was obtained from all the study subjects. The study was conducted in compliance with the ethical guidelines of the 1975 Declaration of Helsinki.

### 2.2. Study Population

A total of 300 age and gender-matched individuals (n = 200 MI cases and n = 100 healthy controls) of Pashtun ethnicity belonging from different districts such as Peshawar, Mardan, Swabi, Charsadda, Nowshehra, Swat, and others of Khyber Pakhtunkhwa were included in the study. The study period was from July 2018 to July 2019. The mean age of the control subjects was 58.43 ± 12.65 (140 males and 60 females), and the control was 56.63 ± 11.87(63 males and 37 females). The diagnosis of MI was based on the American College of Cardiology/American Heart Association (ACC/AHA) classification. A senior cardiologist diagnosed MI based on medical records that revealed medical indications, abnormal cardiac enzymes, ECG (Electrocardiogram) abnormalities, and angiography/echocardiography results. CAD was defined as stenosis ˃50% in at least one of the significant segments of the coronary artery. The control subjects had no lumen stenosis (˂50%) on coronary angiography or physical indications of cardiovascular disease. HTN was defined as having a mean blood pressure of ≥140/90 mmHg or being currently treated for it. DM was classified as having fasting glucose levels of ≥126 mg/dL or non-fasting glucose levels of ≥200 mg/dL, as well as being on oral hypoglycemic medicines or insulin. Patients were admitted tothe three tertiary care (teaching) hospitals of Khyber Pakhtunkhwa, Lady Reading Hospital (LRH) Peshawar, Hayatabad Medical Complex (HMC) Peshawar, and Khyber Teaching Hospital (KTH) Peshawar, while control samples were collected from different districts. Healthy volunteers had no history of cardiovascular disease, especially MI. Inclusion criteria for cases were (i) confirmed MI patients, (ii) Patients belonging to Pakistani Pashtun origin (iii) age ≥30 years. Exclusion criteria were (i) Age ˃80 and ˂30, (ii) mentally ill patients, (iii) severe liver diseases, (iv) malignant tumor, and (v) renal dysfunction. The consent form and thorough demographic, family, and clinical history of all the participants was taken on a carefully designed Proforma. Demographic information includes age, weight, height, and residence. A family history questionnaire includes information on any CVD, MI, or other cardiac issues in the family. The clinical history section of the Proforma includes details about the current disease, co-morbid disorders, and vital signs. For illiterate participants, who have difficulty understanding English, the consent form for their understanding was read and explained in the local Pashtu language and then signed on his/her behalf by any of his/her relatives/attendants. 

### 2.3. Blood Sampling

Following an overnight fast, blood samples were collected from each research participant through venipuncture, with 2.5 mL collected in each EDTA (Ethylene diamine tetra acetic acid) tube and plain tube (without anticoagulant). After allowing the blood in the plain tube to clot, it was centrifuged to obtain serum for biochemical examination. Following aseptic procedures, blood samples (properly labeled) were stored at −10 °C.

### 2.4. DNA Extraction and Biochemical Measurements

Genomic DNA (Deoxyribonucleic acid) was extracted from peripheral blood leukocytes using the WizPrep DNA extraction kit (WizPrep no. W54100). DNA measurements were carried out with the Qubit ™ dsDNA HS Assay kit (Catalog No. Q32851), and the concentration was adjusted to 10 ng/μL.The serum concentration of Total Cholesterol (TC), Triglycerides (TG), LDL-C, and high-density lipoprotein cholesterol (HDL-C) were measured by standard enzymatic methods using standard reagents on Architect Plus (Ci-4100, Germany) biochemical instrument following strictly manufacturer’s instructions in Hospital clinical laboratory.

### 2.5. DNA Samples Pooling

According to the DNA-pooling techniques previously described [37], DNA pools were created from 200 MI patients and 100 control participants in order to cut costs and streamline the sequencing procedure. Each pool contains an equal quantity of genomic DNA (10 ng) from each subject.

### 2.6. Variant Prioritization

The annotated data in the Excel file were first manually curated to screen exonic, and missense variants and synonymous variants were eliminated as shown in Figure 1. The functional influence, biological action, and pathogenicity of the selected variants (SNPs) were checked by using prediction algorithms (PolyPhen and SIFT prediction) built within ANNOVAR.

### 2.7. Validation Trial and Genotyping of APOE and PON1

In the research population, Whole Exome Sequencing (WES) discovered a total of 12 variations in the *APOE* and 16 in the *PON1* gene, respectively. The selected SNPs were genotyped to validate WES results and confirm the association with MI. Sequenom MassARRAY (Sequenom Inch., San Diego, CA, USA) platform was employed following the manufacturer’s instructions.

### 2.8. Statistical Analysis

The SPSS (Statistical Package for the Social Sciences) software was used to analyze statistical data. Age, gender, weight, smoking, lifestyle, exercise, *PON1,* and *APOE* gene variations were the main factors chosen for the study. W Shapiro-Wilk’s test was used to determine the normality of distribution for quantitative data. Categorical data of the cases and control individuals were reported as percentages and frequencies and analyzed with a Chi-square test., whereas continuous variables were displayed as mean standard ± deviation. Odds ratios (OR) of MI cases for each variant using a binary logistic regression model were estimated with a 95% confidence interval (CI). The difference in genotype and allelic prevalence and correlation between cases and control were assessed independently as well as adjusted for conventional risk factors. Age, gender, smoking, and family history of MI, TC, and LDL-C were included as covariates, as well as all the possible genotypes studied. Binary logistic regression was used to determine if the chosen SNP was associated with MI. A *p* ≤ 0.05 was statistically considered significant.

## 3. Results

### 3.1. Population Characteristics

Co-morbidities and Sociodemographic characteristics of study subjects are described briefly in Table 1 and Table 2. The prevalence of co-morbidities such as HTN (55% vs. 36%) and DM (47.5% vs. 32%) were higher in cases as compared to the control subjects. The majority of the subject cases hada family history of MI (55.5%). Moreover, 80% of CAD cases were taking anti-hyperlipidemic medicines (statins) to maintain their poor lipid profile, due to which CAD patients might show normal values of lipid parameters. Moreover, the majority (70%) had a sedentary lifestyle. Furthermore, most of the male patients (58.5%) were smokers. Almost half of the patients were totally non-compliant withdiet and medicines.

### 3.2. WES Results

WES identified a total of 33,329 exonic SNPs, including 3600 homozygous, 29,729 heterozygous, 31,488 synonymous, 1086 deletion, 68 pathogenic, 3456 missenses, and 460 probably damaging variants. A total of 12 variants were identified in *APOE* and 16 in *PON1,* as shown in Table 3 and Table 4. Detailed WES results are shown in Figure 2.

### 3.3. Genotype and Allele Frequencies of APOE (rs429358 and rs7412) and Their Association with MI

Both the *APOE* gene variants (rs429358 and rs7412) were checked for their association with MI by using logistic regression analysis. The genotypic and allelic distributions of both variants are displayed in Table 5. In our study population, the ɛ3 allele is the most common. The results are in broad agreement with data on the frequency ɛ3 allele globally [38]. A significant difference was observed for the variant genotype ɛ3/ɛ4 [OR (95% CI) = 2.13 (1.32–2.65): *p* = 0.031)] and ɛ4 allele [OR (95% CI) = 2.11 (1.25–2.43): *p* = 0.03)] of *APOE* in MI patients compared to control; Whereas other genotypes (ɛ2/ɛ2, ɛ2/ɛ3, ɛ2/ɛ4, ɛ2/ɛ2, and ɛ3/ɛ3) and allele (ɛ2) showed no statistically significant difference (all *p* > 0.05).

### 3.4. Association of L55M and Q192R Variants of PON1 with MI (SNP×MI)

Both the *PON1* gene variants (L55M and Q192R) were checked for their association with MI by using logistic regression analysis. The genotypic and allelic distributions of both variants are displayed in Table 6. The allele and genotype distribution of *PON1* Q192R was found to be significantly different between the MI cases and control subjects. The frequency of the R allele was found to be significantly higher in the study subjects than in the controls. Moreover, the RR genotype was found more frequently in the MI cases than in the controls (16% vs. 9%). By binary logistic regression analysis, the *Q192R* genotype of the *PON1* gene was found to be significantly associated with MI cases [OR (95% CI), 1.353 (0.959–1.910): *p* = 0.048]. There was no significant difference between the MI cases and controls for the L allele and M allele. The results showed no significant association of the *PON1 L55M* genotypes with MI (*p* ˃ 0.05).

### 3.5. Logistic Regression Analysis for MI in Pashtun Population

Logistic regression analysis was performed to determine independent predictors for MI in the study population. On univariate regression analysis, there was a significantly higher risk of MI in the presence of age, gender, smoking, family history of MI, HTN, DM, rs429358 (e4 allele), and rs662 (R allele). Further multivariable analysis showed that participants with ɛ4 and R alleles of rs429358 and rs662 had a significantly higher risk of MI after adjustment for the established conventional risk factors, as shown in Table 7.

## 4. Discussion

The current study investigated the relationship between *APOE* and *PON1* polymorphism and the risk of MI in Pakistan’s Pashtun ethnic population. The genes were selected for genotype validation due to their prominent association with other ethnicities along with data absence in the study population. The selected variants of APOE (rs7412 and rs429358) were genotyped and validated by MassARRAY to confirm the association with MI. The notable variant among the 12 identified variants of *APOE* was rs429358 (*p*.Cys130Arg), located on the 4th exon of chromosome 19. SIFT and PolyPhen predicted the variant rs429358 as deleterious and probably damaging, respectively. Likewise, another exonic missense SNP reported was rs7412 (*p*.Arg176Cys). SIFT and PolyPhen labeled them deleterious and benign, respectively. Furthermore, a significant association between the ɛ4 allele (rs429358) and the risk of MI has been found in the study population, which is in broad agreement with other ethnic populations [27,39,40]. This association remained significant when adjusted for several MI confounding factors.

The *APOE* gene polymorphisms are associated with many diseases such as dementia, Parkinson’s disease, epilepsy, and CAD [41]. Its association with MI or CAD has been extensively studied in the last two decades, and the ϵ4 allele has been found to have a link with it in many studies [42]. Moreover, the same allele was associated with an increased risk of developing HTN [43]. A large-scale genomic study comprising 32,965 controls and 15,492 cases showed that individuals with the ϵ4 allele had a higher risk for coronary heart disease (CHD) compared to individuals with the ϵ3/ϵ3 genotype [44]. However, another study has shown no association of *APOE* gene polymorphism with the development of CAD in the study on the relationship between *APOE* gene polymorphism and blood lipid and CAD in African Caribbean people [45]. These inconsistencies may be because of regional and ethnic variability. This study found the ɛ3 allele to be the most common isoform of the *APOE* gene accounting for 73% of cases and 81% in controls, respectively, which was consistent with most of the previous studies [40,46]. The findings of our study regarding the frequencies of *APOE* allele are consistent with that of other ethnicities [47,48].

Similarly, this study has also assessed the association of *Q192R* and *L55M* variants. Findings suggested the missense SNP*Q192R* (rs662), located on the short arm of chromosome 7 as significant. The frequency of the RR genotype of *Q192R* was found to be higher in the MI cases compared to the control. The *Q192R* (rs662) polymorphism cases with MI revealed a higher frequency of the R allele compared to the control. Both the SIFT and PolyPhen scores predicted it as pathogenic and damaging, respectively. The second missense, exonic SNP, was *L55M* (rs854560). It was shown tolerable and benign by SIFT and PolyPhen score, respectively, and was found not associated with MI (*p* ˃ 0.05). This finding is supported by other studies [23]. Studies conducted in different ethnic populations have shown interesting results of the association of *Q192R* polymorphism of *PON1* with MI [49]. Many studies have revealed the RR genotype and R allele of *PON Q192R* with susceptibility to MI [23]. A study conducted on the Colombian ethnic population proposed *Q192R* polymorphism of *PON1* as a useful biomarker of CAD [50]. Another study also showed an association of the *PON Q192R* variant with CAD [51]. In line with these findings, a significant association was observed for Q192R with CAD by Liu and colleagues [52]. Similar findings were also found in a Chinese ethnic population, south Indian Tamil, and Asian Indians. [53,54,55] Conversely, many other studies have demonstrated conflicting findings and found no association of *PON1 Q192R* polymorphism with CAD [23]. In particular, a genetic study conducted on 120 CAD and 102 healthy volunteers revealed that *PON1 192R* allele frequency was the same among the cases and control [56]. Furthermore, no link was found between the Q192R polymorphism and CAD in a Turkish population [57]. Similarly no association was observed in Taiwan ethnic population [58].

Furthermore, Sociodemographic analysis of cases and controls revealed a higher incidence of DM and HTN in cases compared to the control. Moreover, the results showed an increased prevalence of MI in males compared to females (70% vs. 30%). Most of the MI patients were smokers compared to controls (58.5% vs. 26%). Furthermore, a family history of MI and other heart diseases was more prevalent in some cases. Physically activity (exercise) was found to be very poor in cases compared to controls (30% vs. 70%).

## 5. Conclusions

The present study has suggested that *APOE* variant rs429358 and *PON1* variant Q192R are associated with MIrisk in the Pashtun population of KP and may be further studied to determine their potential as susceptibility biomarkers for the same.

### Limitations

The small sample size is a limitation of our study; similarly, we did not measure the corresponding protein level to know about the expression of the proteins. Moreover, the study was conducted only on patients of Pashtun ethnicity, so it cannot be generalized to the whole of Pakistan or other ethnic populations.

## Figures and Tables

**Figure 1 genes-14-00687-f001:**
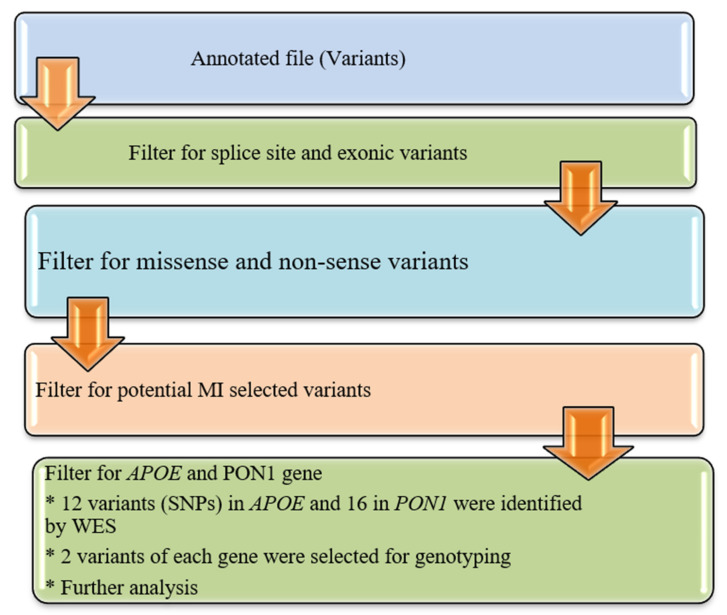
Variants (SNPs) filtration and prioritization pipeline. MI: Myocardial infarction; SNPs: single nucleotide polymorphism; APOE; Apolipoprotein E: PON1; Paroxonase 1.

**Figure 2 genes-14-00687-f002:**
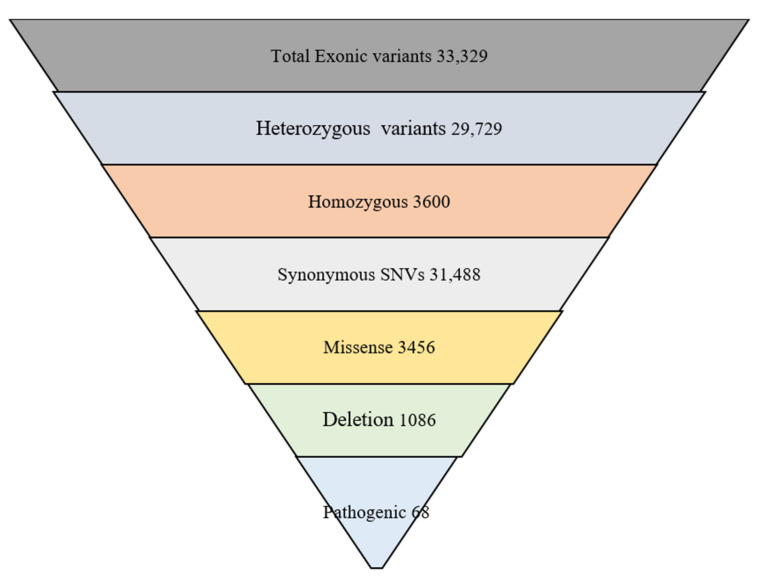
WES results of study subjects.

**Table 1 genes-14-00687-t001:** Co-morbidities in study subjects.

Disease	Frequency (*f*)		*p* Value
	Cases	Control	
HTN	55%	36%	0.001
DM	47.5%	32%	0.007
HCV	0.00%	0.00%	0.000
HBV	0.00%	0.00%	0.000

Abbreviations: HTN: Hypertension; DM: Diabetes mellitus; HCV: Hepatitis C virus; HBV: Hepatitis B virus.

**Table 2 genes-14-00687-t002:** Sociodemographic characteristics of Study subjects.

Variables	Cases n (*f*)	Control n (*f*)	*p*-Value
Gender			0.382
Male	140 (70%)	63 (63%)	
Female	60 (30%)	37(37%)	
Age (yrs)	58.43 ±12.65	56.63 ± 11.87	˂0.001
Age (male)	58.53 ± 11.54	56.32 ± 10.76	
Age (Female)	56.23 ± 12.31	54.12 ± 11.31	
Lipid profile (mg/dL)			
TC	265 ± 15	230 ± 12	˂0.001
HDL-C	58 ± 7	60 ± 5	˂0.001
LDL-C	125 ± 14	100 ± 8	˂0.001
TG	158 ± 10	135 ± 9	0.070
Address			
Peshawar	44 (22%)	28 (28%)	
Charsadda	25(12.5%)	14 (14%)	
Mardan	23 (11.5%)	9 (9%)	
Kohat	27 (13.5%)	10 (10%)	
Swabi	12 (6%)	5 (5%)	0.338
Nowshehra	18 (9%)	5 (5%)	
Bannu	6 (3%)	9 (9%)	
Risalpur	6 (3%)	1 (1%)	
Dir	7(3.5%)	2 (2%)	
Swat	32 (16%)	17 (17%)	
Occupation			
Business	10 (5%)	4 (4%)	
Gov servant	21 (10.5%)	6 (6%)	
Retired	25 (12.5%)	17 (17%)	
Farming	30 (15%)	11 (11%)	˂0.001
Housewife	54 (27%)	59 (59%)	
Labor	60 (30%)	3 (3%)	
Family History of MI			
Yes	111 (55.5%)	21 (21%)	˂0.001
No	89 (44.5%)	79 (79%)	
Exercise			
Yes	60 (30%)	70 (70%)	0.029
No (sedentary)	140 (70%)	30 (30%)	
Smoking			
Yes	117 (58.5%)	26 (26%)	0.096
No	83 (41.5%)	74 (74%)	
Male smokers	58.5%	26 (26%)	˂0.001
Female smokers	0.00%	0.00%	
Diet & Drug Compliance			
Yes	106 (51.5%)	86 (86%)	˂0.001
No	94 (48.5%)	14 (14%)	
Medication history (%)			
Statin	80%	20%	˂0.001
ACEIs	25%	22%	0.542
ARBs	20%	18%	0.482

Abbreviations: n (*f*): number (frequency); yrs: years; TC: total cholesterol; HDL-C: MI: Myocardial Infarction; high-density lipoprotein cholesterol: LDL-C; low-density lipoprotein cholesterol: TG; triglycerides: ACEIs; Angiotensin Converting Enzyme Inhibitors: ARBs; Angiotensin Receptor Blockers.

**Table 3 genes-14-00687-t003:** *APOE* variants (n = 12) identified by WES in Pashtun ethnic population.

SNP ID	Gene	Variant	Chr Position	SIFT Prediction	PolyPhen Prediction	Minor Allele Frequency		Read Depth	
						Cases	Control	Cases	Control
rs769445	*APOE*	C/T	19:44905055	Tol	Benign	0.01	0.03	230	150
rs449647	*APOE*	A/T	19:44905307	Tol	Benign	0.11	0.10	130	110
rs877973	*APOE*	C/A	19:44906026	Tol	Benign	0.08	0.07	280	220
rs184686013	*APOE*	A/G	19:44906286	Tol	Benign	0.04	0.05	170	150
rs429358	*APOE*	T/C	19:44908684	Del	Probably damaging	0.19	0.08	288	150
rs769455	*APOE*	C/T	19:44908783	Tol	Benign	0.03	0.04	157	120
rs7412	*APOE*	C/T	19:44908822	Del	Benign	0.08	0.01	274	140
rs199768005	*APOE*	T/A	19:44909057	Tol	Benign	0.02	0.03	148	130
rs374329439	*APOE*	C/T	19:44909275	Tol	Benign	0.05	0.06	170	140
rs117656888	*APOE*	C/G	19:44909484	Tol	Benign	0.07	0.08	150	170
rs1081105	*APOE*	A/C	19:44909698	Tol	Benign	0.08	0.09	120	150
rs1081106	*APOE*	T/C	19:44910109	Tol	Benign	0.06	0.07	180	220

Abbreviations: SNP; single nucleotide polymorphism: APOE; Apolipoprotein E: Chr; chromosome.

**Table 4 genes-14-00687-t004:** PON1 gene variants (SNPs) were reported by WES in the study population.

SNP ID	Gene	Variant	Chr Position	SIFT Prediction	PolyPhen Prediction	Minor Allele Frequency (%)		Read Depth	
						Cases	Control	Cases	Control
rs372449149	*PON1*	G/A	7:94928371	Tol	Benign	0.23	0.25	234	250
rs185623242	*PON1*	G/A	7:94931521	Tol	Benign	0.30	0.28	140	330
rs369422555	*PON1*	C/G	7:94931583	Tol	Benign	0.34	0.32	546	237
rs371803280	*PON1*	C/T	7:94931624	Tol	Benign	0.45	0.43	120	235
rs370355032	*PON1*	G/A	7:949337419	Tol	Benign	0.27	0.29	543	454
rs80019660	*PON1*	G/A	7:94937419	Tol	Benign	0.12	0.16	453	123
rs662	*PON1*	T/C	7:94937446	Path	damaging	0.40	0.30	543	436
rs61736513	*PON1*	C/T	7:94944679	Tol	Benign	0.44	0.47	124	453
rs371338407	*PON1*	G/C	7:94944768	Tol	Benign	0.33	0.36	123	342
rs8545560	*PON1*	A/T	7:94946084	Tol	Benign	0.08	0.01	154	234
rs149100710	*PON1*	C/T	7:94947635	Tol	Benign	0.47	0.50	452	734
rs144612002	*PON1*	T/C	7:94947638	Tol	Benign	0.38	0.35	563	542
rs138512790	*PON1*	A/G	7:9494756	Tol	Benign	0.43	0.45	745	121
rs141665531	*PON1*	G/A	7:94947661	Tol	Benign	0.31	0.34	234	534
rs146211440	*PON1*	A/C	7:94953721	Tol	Benign	0.45	0.43	523	123
rs150657027	*PON1*	G/A	7:94953771	Tol	Benign	0.23	0.25	534	213

Abbreviations: SNP; single nucleotide polymorphism: PON1; Paroxonase 1: Chr; chromosome.

**Table 5 genes-14-00687-t005:** Genotypes and alleles distribution of *APOE* gene in MI patients and controls.

Genotypes	MI Patients (*n* = 200)	Controls (*n* = 100)	OR (95% CI)	*p* Value
ɛ2/ɛ2	6 (3%)	4 (4%)	0.84 (0.26–2.94)	0.841
ɛ2/ɛ3	20 (10%)	15 (15%)	0.65 (0.32–1.27)	0.207
ɛ2/ɛ4	8 (4%)	6 (6%)	0.72 (0.13–2.86)	0.665
ɛ3/ɛ3	127 (63.5)	67(67%)	1.20 (0.76–2.22)	0.403
ɛ3/ɛ4	27 (13.5%)	3 (3%)	2.13 (1.32–2.65)	0.031
ɛ4/ɛ4	12 (6%)	5 (5%)	0.45 (0.05–4.48)	0.532
Alleles, n				
ɛ2	40	29	0.87 (0.34–2.06)	0.462
ɛ3	174	152	1.70 (1.09–2.23)	0.004
ɛ4	47	19	2.11 (1.25–2.43)	0.030
Allelic carriage rate				
ɛ2 (+)	34 (17%)	25%	ref	-
ɛ2 (−)	166 83%)	75%	0.614 (0.343–1.102)	0.070
ɛ4 (+)	47 (23.5)	14%	ref	-
ɛ4 (−)	153 (76.5)	86%	1.741 (0.919–3.297)	0.057

Note: Odd ratio (95% CI) obtained from binary logistic regression analysis, *p* ≤ 0.05, was considered significant. MI: Myocardial infarction; OR: Odd ratio; CI: Confidence interval.

**Table 6 genes-14-00687-t006:** Genotyping distribution and allele frequencies of *PON1* gene (*Q192R and L55M*) polymorphisms in MI cases and controls.

PON1 Gene Genotypes	MI Cases *n* (*f*)	Controls (*f*)	Association Tests with MI		
			*p* Value	OR	95% CI
*Rs662*	-	-	-	-	-
Total	200 (100)	100	-	-	-
QQ (GG)	72 (36%)	49%	ref	-	-
QR (AG)	96 (48%)	42%	0.166	1.160	0.844–1.523
RR (AA)	32(16%)	9%	0.048	1.523	1.087–2.132
Allele frequency (n)					
Q (G)	240 (60)	70%	ref	-	-
R (A)	160 (40)	30%	0.048	1.353	0.959–1.910
Allelic carriage rate					
Q (+)	168 (84)	51%	ref		
Q (−)	32 (16)	49%	0	5.044	2.926–8.696
R (+)	128 (64)	74%	ref		
R (−)	72 (36)	26%	0.006	0.625	1.24–2.763
*L55M variant polymorphism*					
Total					
LL	54.08 (27.04%)	24.01%	ref	-	-
LM	99.84(49.92%)	49.98%	0.500	1.015	0.798–1.291
MM	46.08 (23.04%)	26.01%	0.357	0.898	0.592–1.363
Allele frequency (n)					
L	208	98	ref	-	-
M	192.03	102	0.402	0.955	0.751–1.216
Allelic carriage rate					
L (+)	153.92	73.99	ref		
L (−)	46.08	26.01	0.058	1.556	0.931–2.598
M (+)	145.92	75.99	ref		
M (−)	54.08	24.01	0.579	0.833	0.478–1.449

Abbreviations: MI: Myocardial infarction; *PON1*: Paroxonase1; Q: glutamine; R: arginine; L: leucine; M, methionine, OD: Odd ratio; CI: confidence interval (A *p* ≤ 0.05 was statistically considered significant).

**Table 7 genes-14-00687-t007:** Logistic regression analysis of the risk of MI in the Pashtun population.

Variable	Crude Values			Adjusted Values		
	β-Coefficient	*p* Value	OR (95% CI)	β-Coefficient	*p* Value	OR (95% CI)
Age	0.051	˂0.001	1.04 (1.21–1.78)	0.042	˂0.001	1.03 (1.04–1.85)
Gender	0.612	0.382	0.85 (0.43–0.75)	0.542	0.257	0.272 (0.41–0.68)
Smoking	1.243	˂0.001	1.75 (1.32–1.76)	1.121	˂0.001	1.56 (1.27–1.58)
Family history of MI	0.032	˂0.001	1.25 (1.12–2.07)	0.21	˂0.001	1.04 (1.11–1.98)
HTN	1.132	˂0.001	2.12 (1.42–1.93)	1.032	˂0.001	2.01 (1.25–1.87)
DM	1.545	0.007	1.43 (1.65–2.01)	1.423	0.005	1.36 (1.54–1.98)
TC	0.063	˂0.001	1.73 (1.65–2.32)	0.051	˂0.001	1.71 (1.67–2.32)
LDL-C	0.045	˂0.001	1.54 (1.32–2.15)	0.032	˂0.001	1.45 (1.25–2.01)
ɛ4 allele	0.422	0.030	2.11 (1.25–2.43)	0.321	0.024	1.98 (1.16–2.36)
R allele	0.362	0.048	1.35 (0.95–1.91)	0.234	0.037	1.25 (1.22–2.43)

Note: OR: Odd ratio; CI: Confidence interval; MI: Myocardial infarction: HTN: Hypertension; DM: Diabetes mellitus; TC: Total cholesterol; LDL-C: Low-density lipoprotein cholesterol. Adjusted indicates results adjusted for covaries, i.e., age, gender, smoking, family history of MI, DM, HTN, and lipid parameters.

## Data Availability

All data is available with manuscript.

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
