# Peer review of "Association of APOE (rs429358 and rs7412) and PON1 (Q192R and L55M) Variants with Myocardial Infarction in the Pashtun Ethnic Population of Khyber Pakhtunkhwa, Pakistan"

_genes, 2023, doi:10.3390/genes14030687_

Round 1
Reviewer 1 Report
The authors studied the relationship between APOE and PON1gene variants in CAD patients and healthy subjects in the Pashtun ethnicity of Pakistan. They found several APOE variants, rs429358 and Q192R (rs662) was, significantly associated with cases.
The main message needs to be more specific. There are concerns and suggestions that the authors may want to address.
1) The authors need to quote the relevant references (ex: reference 8). Also many places, the references are missing (ex: sociodemographic references are missing).
2) In the introduction authors state that “parental history of 61 premature atherosclerosis has a 1.5- to 2-fold risk” They might want to check if the fold change provided is correct.
3) Inclusion and exclusion criteria need to be detailed. They have given an exclusion age of 80 years. What is the lower cutoff?
4) There is no mention of the exclusion of renal dysfunction subjects or any major concomitant infection, as they are one of the major risk factors of CAD.
5) How were the presence of CAD and cardiovascular events verified?
6) Needs to provide more details of the statistical analysis. Tools used for the analysis; tests used for categorical, and continuous variables. Did they check for normality how did they deal with non-normal data?
7) What does f stand for in table 2.
8) Age needs to be provided in the Demographic details, Age for males and females needs to be given. is the study age and gender-matched?
9) How did they measure biochemical parameters?
10) samples collection time point? What percent of these patients are on medication?
11) What does UA stand for?
12) While all the lipid parameters are presented in mg/L, why is only UA is presented in mmol/L? The unit and values provided make no sense; they need to check the units given for each biochemical parameter.
13) Need to include legends for all the tables.
14) What is the importance of p<0.07?
15) While authors have found a significant association with the APOE and PON1 variants, the ORs for individual allele variant needs to be calculated. The model built should then be adjusted to conventional risk factors. ROC analysis needs to be included.
16) Based on the current analysis proposing the two variants, a potential genetic biomarker for MI is an overstatement.
Reviewer 2 Report
The authors investigated the association of APOE and PON1 with MI n the Pashtun ethnic 37 population of Pakistan. The authors conducted a lot of work in clarifying the role of these variants but the manuscript needs improving. the reviewer has the following comments:
1. The main issue is how the authors select cases and controls? Why the controls is only half of the cases? Did you conduct power/sample size calculation?
2. Line 34: missing space
3. Line 89: extra space
4. Line 184 185 199 200 210 225: not sure what happened but showed “Error! Reference source 184 not found”
5. Footnotes are missing for all the Tables.
Round 2
Reviewer 1 Report
It is unclear how the authors verified the CAD or the controls. Additional information regarding their verification process is necessary.
The methodology of the lipid profiling needs to be outlined in detail.
How was the presence of DM assessed? The authors can provide fasting glucose and HbA1C values to clarify.
The statistical significance of smoking status should be evaluated separately for males and females to account for potential gender differences.
What is Diet & Drug Compliance? The authors should incorporate medication history separately to provide a clearer understanding of this factor's impact.
The LDL-c data for the cases appears lower; it is unclear what percentage of the patient population is on statins, and whether LDL was measured using a biochemical assay or Friedewald’s formula. Similarly, the LDL levels in the healthy controls seem surprisingly lower. It would be helpful if the authors to stratify the data based on gender to provide a more nuanced analysis.
Gender should be used for statistical adjustment to account for any potential gender-specific effects.
It would be valuable for the authors to conduct a ROC analysis to evaluate the efficacy of their findings.
Reviewer 2 Report
I don't have further comments.
